# WebDancer:
# Towards Autonomous Information Seeking Agency

**Jialong Wu**[*][¶], **Baixuan Li**[*], **Runnan Fang**[*], **Wenbiao Yin**[*][¶], **Liwen Zhang,**
**Zhenglin Wang, Zhengwei Tao, Dingchu Zhang, Zekun Xi, Xiangru Tang,**
**Yong Jiang**[¶]**, Pengjun Xie, Fei Huang, Jingren Zhou**

Tongyi Lab 🔷 , Alibaba Group
✉ Correspondence to: `wujialongml@gmail.com`
`{yinwenbiao.ywb,yongjiang.jy}@alibaba-inc.com`

## Abstract

Addressing intricate real-world problems necessitates in-depth information seeking and multi-step reasoning. Recent progress in agentic systems, exemplified by *Deep Research*, underscores the potential for autonomous multi-step research. In this work, we present a cohesive paradigm for building end-to-end agentic information seeking agents from a data-centric and training-stage perspective. Our approach consists of four key stages: (1) agentic data synthesis, (2) trajectory sampling, (3) supervised fine-tuning for effective cold start, and (4) agentic reinforcement learning for enhanced generalization. We instantiate this framework in a web agent based on the `ReAct`, **WebDancer**. Empirical evaluations on the challenging information-seeking benchmarks, GAIA and WebWalkerQA, demonstrate the strong performance of WebDancer, achieving considerable results and highlighting the efficacy of our training paradigm. Further analysis of agent training provides valuable insights and actionable, systematic pathways for developing more capable agentic models. [2]

## 1 Introduction

Web agents are autonomous systems that perceive their real-world web environment, make decisions, and take actions to accomplish specific and human-like tasks. Recent systems, such as ChatGPT *Deep Research* [1] and Grok *DeepSearch* [2], have demonstrated strong deep information-seeking capabilities through end-to-end reinforcement learning (RL) training.

The community's previous approaches for information seeking by agentic systems can be categorized into two types: *(i)* Directly leveraging prompting engineering techniques to guide Large Language Models (LLMs) or Large Reasoning Models (LRMs) [3–5] to execute complex tasks. *(ii)* Incorporating *search* or *browser* capabilities into the web agents through supervised fine-tuning (SFT) or RL [6, 5, 7–10]. The first training-free methods are unable to effectively leverage the reasoning capabilities enabled by the reasoning model. Although the latter methods internalize certain information-seeking capabilities through SFT or RL training, both the training and evaluation datasets are relatively simple and do not capture the real-world challenges, for instance, performance on the 2Wiki dataset has already reached over 80%. Moreover, the current SFT or RL training paradigm does not fully and efficiently exploit the potential of information-seeking behavior. Building autonomous information seeking agency involves addressing a set of challenges that span web environment perception and decision-making: (1) acquiring high-quality, fine-grained browsing data that reflects

---

[*]Equal contribution. [¶] Correspondence.

[2]The codes and demo are released in `https://github.com/Alibaba-NLP/DeepResearch`.

diverse user intents and rich interaction contexts, (2) constructing reliable trajectories that support long-horizon reasoning and task decomposition, and (3) designing scalable and generalizable training strategies capable of endowing the web agent with robust behavior across out-of-distribution web environments, complex interaction patterns, and long-term objectives.

To address these challenges, our objective is to **unlock the autonomous multi-turn information-seeking agency, exploring how to build a web agent like *Deep Research* from scratch.** An agent model like *Deep Research* produces sequences of interleaved reasoning and action steps, where each action invokes a tool to interact with the external environment **autonomously**. Observations from these interactions guide subsequent reasoning and actions until the task is completed. This process is optimized through end-to-end tool-augmented training. The ReAct framework [11] is the most suitable paradigm, as it tightly couples reasoning with action to facilitate effective learning and generalization in interactive settings.

We aim to provide the research community with a systematic guideline for building such agents from a *data-centric* and *training-stage* perspective.

From a *data-centric* perspective, constructing web QA data is crucial to building web agents, regardless of whether the training paradigm is SFT or RL. Widely used QA datasets are often shallow, typically consisting of problems that can be solved with a single or a few-turn search. Previous works often filter the difficult QA pairs from open-sourced human-labeled datasets using prompting techniques [7]. Additionally, challenging web-based QA datasets typically only have test or validation sets, and their data size is relatively small. For example, GAIA [12] only has 466, WebWalkerQA [3] contains 680 examples, and BrowseComp [13] has 1,266, making them insufficient for effective training. Therefore, the automatic synthesis of high-quality datasets becomes crucial. [14, 15]. We synthesise the datasets in two ways: 1). By crawling web pages to construct deep queries, referred to as **CRAWLQA**, enabling the acquisition of web information through click actions. 2). By enhancing *easy-to-hard* QA pairs synthesis to incentivize the progression from *weak-to-strong* agency, transforming simple questions into complex ones, termed **E2HQA**.

From a *training-stage* perspective, prior work has explored SFT or off-policy RL, but these approaches often face generalization issues, particularly in complex, real-world search environments. Other methods adopt on-policy RL directly [6], but in multi-tool settings, early training steps tend to focus primarily on learning tool usage via instruction following. To enable more efficient and effective training, we adopt a two-stage approach combining rejection sampling fine-tuning (RFT) with subsequent on-policy RL. For the trajectory sampling, we restrict the action space to two commonly effective web information-seeking tools as *action*: *search* 🔍 and click 👆 . Building on this setup, we employ rejection sampling to generate trajectories using two prompting strategies: one with a strong instruction LLMs for Short-CoT and another leveraging the LRMs for Long-CoT. These yield high-quality trajectories containing either short or long thought, respectively. In the RL stage, we adopt the Decoupled Clip and Dynamic Sampling Policy Optimization (DAPO) algorithm [16], whose *dynamic sampling* mechanism can effectively exploit QA pairs that remain underutilized during the SFT phase, thereby enhancing data efficiency and policy robustness.

**Our key contributions can be summarized as follows: we abstract the end-to-end web agents building pipeline into four key stages**: **Step** I: Construct diverse and challenging deep information seeking QA pairs based on the real-world web environment (§2.1); **Step** II: Sample high-quality trajectories from QA pairs using both LLMs and LRMs to guide the agency learning process (§2.2); **Step** III: Perform fine-tuning to adapt the format instruction following to agentic tasks and environments (§3.1); **Step** IV: Apply RL to optimize the agent's decision-making and generalization capabilities in real-world web environments (§3.2). We offer a systematic, end-to-end pipeline for building long-term information-seeking web agents.

Extensive experiments on two web information seeking benchmarks, GAIA and WebWalkerQA, show the effectiveness of our pipeline and WebDancer (§4). We further present a comprehensive analysis covering data efficiency, agentic system evaluation, and agent learning (§5).

WebDancer represents the **first end-to-end trained deep research agent** within the Tongyi DeepResearch series [17]. It is among the earliest works in the community to propose a **fully end-to-end training pipeline** for deep research agents, pioneering studies on both **agentic data synthesis** for deep research tasks and **agentic reinforcement learning**.

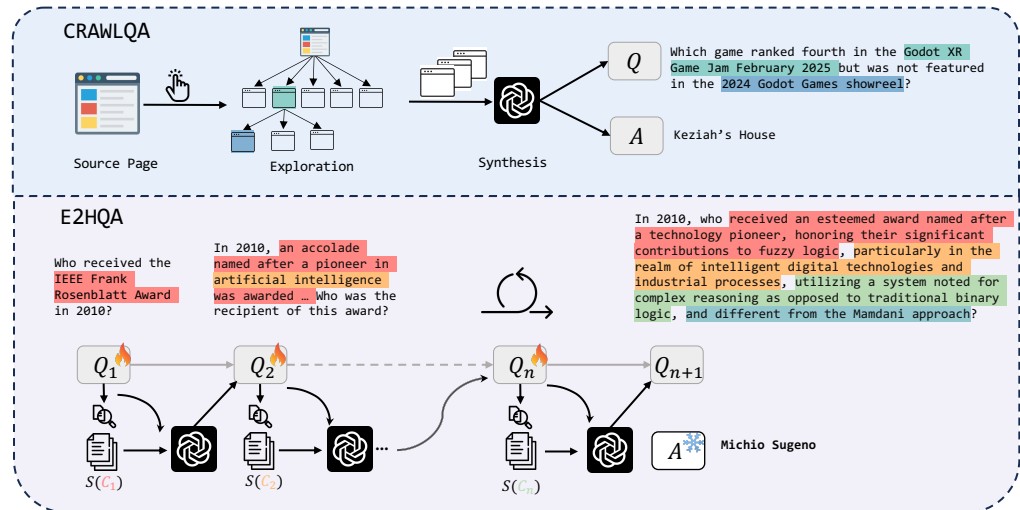

Figure 1: **Two web data generation pipelines.** ❶ For **CRAWLQA**, we first collect `root` url of knowlageable websites. Then we mimic human behavior by systematically clicking and collecting subpages accessible through sublinks on the `root/...` page. Using predefined rules, we leverage GPT4o to generate synthetic QA pairs based on the gathered information. ❷ For **E2HQA**, the initial question $Q_1$ is iteratively evolved using the new information $C_i$ retrieved from the entity $E_i$ at iteration $i$, allowing the task to progressively scale in complexity, from simpler instances to more challenging ones. We use GPT-4o to rewrite the question until the iteration reaches $n$.

## 2 Deep Information Seeking Dataset Synthesis

### 2.1 QA Pairs Construction

To enable longer-horizon web exploration trajectories, it is essential to curate a substantial corpus of complex and diverse QA pairs that can elicit multi-step reasoning, goal decomposition, and rich interaction sequences. The main requirements for these QAs are: *(i)* diversity of question types, and *(ii)* increased task complexity as measured by the number of interaction steps required for resolution. In contrast to prior datasets that predominantly involve shallow queries solvable in 2–3 steps, our objective is to scale both the volume and the depth of multi-hop reasoning. To achieve this, we primarily develop the below datasets: **CRAWLQA** and **E2HQA**.

**CRAWLQA**  Constructing QA pairs based on information crawled from web pages represents an effective paradigm for scalable knowledge acquisition [3]. We begin by collecting the root URLs of official and knowledgeable websites spanning arxiv, github, wiki, *etc.* [12] To emulate human browsing behavior, we recursively navigate subpages by following accessible hyperlinks from each root site. We employ GPT-4o to synthesize QA pairs from the collected content. To ensure specificity and relevance of questions, inspired by Sen et al. [18], we prompt LLMs to generate questions of designed types (*e.g.*, COUNT, MULTI-HOP, INTERSECTION) via in-context learning [19].

**E2HQA**  Similar to the reverse construction strategy [13, 20], we begin from large QA pairs in SimpleQA style [21] where each answer is a concise, fact-seeking entity. We first select an entity $E_n$ from the question $Q_n$, where $n$ represents the number of refinement iterations. Then, we use the LLMs to construct a query based on this entity in order to search via search engine $S$ for information $C_n$ related to $E_n$. After that, we use LLMs $\pi$ to restructure the obtained content into a new query $R_n$ to replace the original entity in the question. The process can be signaled as: $R_n = \pi(S(C_n))$. This way, the new question $Q_{n+1}$ requires solving the sub-problem we have constructed before finding the answer to the original question. Moreover, it ensures that the answer does not change during the question refinement, thereby preserving the validity of the QA pairs. By continuously searching, we can gradually rephrase an initially simple question into a more complex multi-step one. Moreover, the number of steps needed to solve the problem can be controlled by adjusting the number of rephrasing times.

## 2.2 Agent Trajectories Rejection Sampling

**Agent Setup**   Our agent framework is based on `ReAct` [11], the most popular approach to language agents. A `ReAct` trajectory consists of multiple `Thought`-`Action`-`Observation` rounds, where an LM generates free-form `Thought` for versatile purposes, and structured `Action` to interact with environments (tools) and receive `Observation` feedback. We assume that the agent execution loop at time $t$ can be denoted as $(\tau_t, \alpha_t, o_t)$, where $\tau$ denotes `Thought`, $\alpha$ signifies `Action`, and $o$ represents `Observation`. $\alpha$ can be further expressed as $(\alpha^m, \alpha^p)$, where $\alpha^m$ is the name of the action, and $\alpha^p$ is the parameters required to perform the action. $\alpha^m \in \{search, visit, answer\}$, which corresponds to the two most important agentic tools in the deep information seeking. For *search* `action`, $\alpha^p$ consists of *query* and *filter_year*, while for *visit* `action`, $\alpha^p$ consists of *goal* and *url_link*. The `observation` of *search* `action` includes the Top-10 titles and snippets, whereas the `observation` of the *visit* `action` is the *evidence* and *summary*, generated by a summarizer model $M_s$. The iteration terminates when the `action` is *answer*.

Then the historical trajectory can be signaled as:

$$\mathcal{H}_t = (\tau_0, \alpha_0, o_0, \tau_1, ..., \tau_{t-1}, \alpha_{t-1}, o_{t-1}). \tag{1}$$

At time step $t$, the agent receives an `observation` $o_t$ from the web environment and generates `thought` $\tau_t$ taking an `action` $\alpha_t$, following poliy $\pi(\tau_t, \alpha_t | \mathcal{H}_t)$.

The Chain-of-Thought (CoT) method has significantly enhanced the inferential capabilities of LLMs through a step-by-step reasoning process [22], corresponding to the `thought` component in agentic systems. This process is critical for agentic execution, enabling high-level workflow planning, self-reflection, information extraction, adaptive action planning, and accurate `action` (tool usage).

**Short and Long CoT Construction**   Agent models internalise the CoT generation capability as an active behavioral component of the model. [23, 24] The length of CoT and the associated thinking patterns play a crucial role in performance [25–27] We propose two simple yet effective methods for constructing the short CoT and long CoT, respectively. For short CoTs, we directly leverage the `ReAct` framework to collect the trajectories using a powerful model, GPT-4o. For long CoTs, we sequentially provide the LRMs, QwQ-Plus, with the historical `actions` and `observations` at each step, enabling it to decide the next `action` autonomously. Notably, we **exclude** the previous `thought` during further inference, as the LRM, QwQ-Plus, has not been exposed to multi-step reasoning inputs during training. However, we retain the `thought` at each step in the generated trajectory, as they serve as valuable supervision signals. The LRM's intermediate reasoning process, denoted as, denoted as "*<reasoning_content>*", is recorded as the current `thought` of the current step. Each constructed QA instance undergoes rejection sampling up to $N$ times to ensure quality and coherence.

**Trajectories Filtering**   We adopt a three-stage funnel-based trajectory filtering framework consisting of *validity control*, *correctness verification*, and *quality assessment*.

- For *validity control*, directly prompting LLMs to generate responses in the `ReAct` format under long-content conditions may result in non-compliance with instructions. In such cases, we discard these data points.
- For *correctness verification*, we only retain correct results. We follow the evaluation methodology proposed by Phan et al. [28] and Wei et al. [13] and use GPT-4o for accurate judgment.
- For *quality assessment*, we first apply rules to filter out trajectories with more than two actions, ensuring that there are no hallucinations and no severe repetitions. Subsequently, we filter the trajectories based on prompting to retain those that meet the following three criteria: Information Non-redundancy, Goal Alignment, and Logical Reasoning and Accuracy.

The QA pairs that are not present in the SFT dataset can be utilized during the reinforcement learning stage effectively. [3]

## 3   Multi-Step Multi-Tool Agent Learning

After obtaining high-quality trajectories in `ReAct` format, we seamlessly incorporate them into our agent SFT training stage. Specifically, `Thought` segments are closed by `<think>` and `</think>`, `Action`

---

[3]The details of training datasets and are shown in App. D.

segments by <tool_call> and </tool_call>, `Observation` segments by <tool_response> and </tool_response>. The final `Action` segment corresponds to the final answer, enclosed by <answer> and </answer>. In addition, the QA data without trajectories, which those filtered during earlier stages, can be effectively leveraged during the RL phase. We first train a policy model $\pi_\theta$ via agent SFT for cold start, followed by agent RL for generalization. The overall training framework is illustrated in Figure 2.

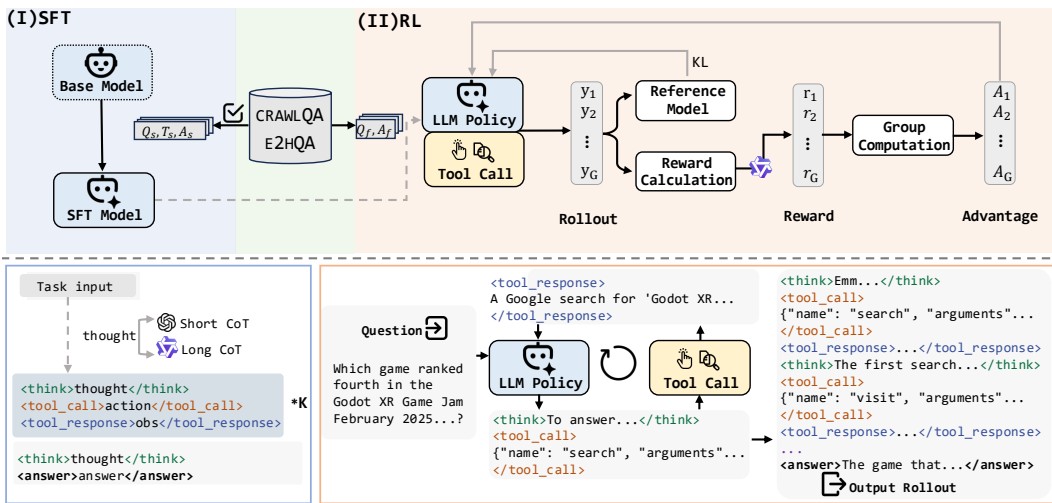

Figure 2: **The overview of training framework**. (I) The SFT stage for cold start utilizes the reformatted `ReAct` datasets, where the thought includes both short and long CoT, respectively. (II) The RL stage performs rollouts with the tool calls on the QA pairs that are not utilized during the SFT stage, and optimizes the policy using the DAPO algorithm.

## 3.1 Agent Supervised Fine Tuning

To capture complete agentic trajectories, we train the policy model $\theta$ via supervised fine-tuning on obtained decision-making trajectories. The cold start enhances the model's capability to couple multiple reasoning and action steps, teaching it a behavioral paradigm of alternating reasoning with action, while preserving its original reasoning capabilities as much as possible. Following the empirical findings of Chen et al. [29, 6], Zhang et al. [23], to avoid interference from external feedback during learning, we mask out loss contributions from `observation` in the agentic world modelling task, which has been shown to generally improve performance and robustness. Given the task context **tc** and the complete agentic execution trajectory $\mathcal{H} = (x_0, x_1, ..., x_{n-1}, x_n)$, where each $x_i \in \{\tau, \alpha, o\}$, the loss function $L$ is computed as follows:

$$L = -\frac{1}{\sum_{i=1}^{|\mathcal{H}|} \mathbb{I}[x_i \neq o]} \sum_{i=1}^{|\mathcal{H}|} \mathbb{I}[x_i \neq o] \cdot \log \pi_\theta(x_i \mid \mathbf{tc}, x_{<i}) \tag{2}$$

Here, $\mathbb{I}[x_i \neq o]$ filters out tokens corresponding to external feedback, ensuring that the loss is computed over the agent's autonomous decision steps. The SFT stage offers strong initialization for the subsequent RL stage [30].

## 3.2 Agent Reinforcement Learning

The agent RL stage aims to internalize the agency capability into the reasoning model, enhancing its multi-turn, multi-tool usage capacity with outcome-based rewards. [31] Building on the SFT stage, RL employs Decoupled Clip and Dynamic Sampling Policy Optimization algorithm to refine and incentivize the policy model $\pi_\theta$'s ability to interleave `Thought`-`Action`-`Observation` sequences. **DAPO** Decoupled Clip and Dynamic Sampling Policy Optimization (**DAPO**) algorithm is an RL algorithm that optimizes a policy $\pi_\theta$ to produce higher-reward outputs under a reward model $R$ [16, 32]. For each question-answer pair $(q, a)$ from the data distribution $\mathcal{D}$, DAPO samples a set of candidate agentic executions $\{o_i\}_{i=1}^G$. The policy is then updated to maximize the following objective:

$$\mathcal{J}_{\text{DAPO}}(\theta) = \quad \mathbb{E}_{(q,a)\sim\mathcal{D},\{o_i\}_{i=1}^{G}\sim\pi_{\theta_{\text{old}}}(\cdot|context)}$$

$$\left[ \frac{1}{\sum_{i=1}^{G}|o_i|} \sum_{i=1}^{G} \sum_{t=1}^{|o_i|} \min\left( r_{i,t}(\theta)\hat{A}_{i,t}, \text{ clip}\left(r_{i,t}(\theta), 1-\varepsilon_{low}, 1+\varepsilon_{high}\right)\hat{A}_{i,t}\right)\right]$$

$$\text{s.t.} \quad 0 < \left| \{o_i \mid \textbf{is\_equivalent}(y, o_i)\}\right| < G, \tag{3}$$

where agentic execution $o_i$ refers solely to the tokens generated by models, excluding any tool responses. In contrast, $context$, including both the model outputs and tool responses, is used to construct the input trajectory for computing $\pi_{\theta_{\text{old}}}$. However, the optimization is applied only to the model-generated portion $o_i$, aligning with the SFT. $\varepsilon$ is the clipping range of the importance sampling ratio $r_{i,t}(\theta)$. And $\hat{A}_{i,t}$ is an estimator of the advantage of the $i$-th agentic executions at time step $t$:

$$r_{i,j}(\theta) = \frac{\pi_\theta\left(o_i \mid q_i, o_{i,<t}\right)}{\pi_{\theta_{\text{old}}}\left(o_i \mid q_i, o_{i,<t}\right)}, \quad \hat{A}_{i,j} = \frac{R_i - \text{mean}\left(\{R_i\}\right)}{\text{std}\left(\{R_i\}\right)}, \tag{4}$$

The dynamic sampling mechanism over-samples and filters out prompts with accuracy equal to 1 and 0. It is crucial in our data-training pipeline, as the remaining QA pairs, being synthetically generated—may contain invalid or noisy instances that could otherwise degrade policy learning. Such unreliable samples can be effectively ignored, ensuring the agent focuses on learning from high-quality signals.

**Agentic Action Rollout** Within the `ReAct` framework, each round of agentic execution begins by generating a `thought`, closed by `<think>` and `</think>`, followed by a `action` name $\alpha^m$ and corresponding parameters $\alpha^p$, enclosed by `<tool_call>` and `</tool_call>` operation, all conditioned on the iteration history $\mathcal{H}$. These components are iteratively used to interact with the real-world search environment, producing an `observation` as feedback, bounded by `<tool_response>` and `</tool_response>` upon the `<tool_response>` is detected. The round of interaction spans from `<think>` to `</tool_response>`. The rollout concludes with the generation of `<answer>` and `</answer>`, following the final `thought`.

**Reward Design** The reward design plays a critical role during the RL training process [26]. Our reward system mainly consists of two types of rewards, $score_{\text{format}}$ and $score_{\text{answer}}$. Given that format consistency has been largely addressed during the initial RFT stage, we assign a small weight to the $score_{\text{format}}$ in the overall reward. The $score_{\text{format}}$ is binary: it is set to 1 only if the entire output strictly conforms to the required format and all tool calls in *json* format are valid. Considering that the QA answers are inherently non-verifiable, cannot be reliably evaluated using rule-based F1/EM metrics, despite the brevity of the responses, and that the final evaluation relies on *LLM-as-Judge* [33] which the judge model is $M_j$, we opt to employ model-based prompt evaluation as the answer reward signal [34–36]. The $score_{\text{answer}}$ is also binary, assigned as 1 only when the response is judged as correct by the LLMs. The final reward function is:

$$R(\hat{y}_i, y) = 0.1 * score_{\text{format}} + 0.9 * score_{\text{answer}} \tag{5}$$

where $\hat{y}_i$ denotes the model prediction and $y$ is the reference answer.

## 4 Experiments

### 4.1 Experimental Setup

We evaluate our approach on two established deep information-seeking benchmarks: **GAIA** and **WebWalkerQA**. In this work, we adopt the *LLM-as-Judges* paradigm to evaluate both tasks using the `Pass@1` metric, following [4]. The details of the datasets and baselines are introduced in App. E.1 and App. E.2, respectively. The implementation details are shown in App. E.3. Qwen-7B and Qwen-32B are trained on Short-CoT datasets, while QwQ-32B is trained on Long-CoT datasets. Further analyses are shown in Sec. 5.

### 4.2 Experimental Results

**Main Results** As shown in Table 1, frameworks without agentic capabilities (*No Agency*) perform poorly on both the GAIA and WebWalkerQA benchmarks, highlighting the necessity of active

Table 1: **Main results** on GAIA and WebWalkerQA benchmarks. We discuss the reported results of baselines and concurrent works in App. C.1. "-" means results that are either not reproducible or not reported. The best results among all frameworks are in **bolded**.

| Backbone | Framework | GAIA | | | | WebWalkerQA | | | |
|---|---|---|---|---|---|---|---|---|---|
| | | Level 1 | Level 2 | Level 3 | Avg. | Easy | Medium | Hard | Avg. |
| *No Agency* | | | | | | | | | |
| Qwen-2.5-7B | Base | 12.8 | 3.8 | 0.0 | 6.8 | 1.25 | 0.8 | 0.7 | 0.8 |
| Qwen-2.5-32B | Base | 20.5 | 9.6 | 8.3 | 13.6 | 3.8 | 2.5 | 3.3 | 3.1 |
| | RAG | 12.8 | 11.8 | 8.3 | 11.8 | 23.1 | 14.3 | 11.3 | 15.3 |
| Qwen-2.5-72B | Base | 20.5 | 13.5 | 0.0 | 14.6 | 9.4 | 7.1 | 3.3 | 6.3 |
| GPT-4o | Base | 23.1 | 15.4 | 8.3 | 17.5 | 6.7 | 6.0 | 4.2 | 5.5 |
| QwQ-32B | Base | 30.8 | 15.4 | 25.0 | 22.3 | 7.5 | 2.1 | 4.6 | 4.3 |
| | RAG | 33.3 | 36.5 | 8.3 | 32.0 | 36.9 | 26.1 | 33.5 | 31.2 |
| DeepSeek-R1-671B | Base | 43.6 | 26.9 | 8.3 | 31.1 | 5.0 | 11.8 | 11.3 | 10.0 |
| *Close-Sourced Agentic Frameworks* | | | | | | | | | |
| | *OpenAI DR* | 74.3 | 69.1 | 47.6 | 67.4 | - | - | - | - |
| *Open-sourced Agentic Frameworks* | | | | | | | | | |
| Qwen-2.5-7B | Search-o1 | 23.1 | 17.3 | 0.0 | 17.5 | - | - | - | - |
| | R1-Searcher | 28.2 | 19.2 | 8.3 | 20.4 | - | - | - | - |
| Qwen-2.5-32B | Search-o1 | 33.3 | 25.0 | 0.0 | 28.2 | - | - | - | - |
| QwQ-32B | Search-o1 | 53.8 | 34.6 | 16.7 | 39.8 | 43.1 | 35.0 | 27.1 | 34.1 |
| | WebThinker-Base | 53.8 | 44.2 | 16.7 | 44.7 | 47.2 | 41.1 | 39.2 | 41.9 |
| | WebThinker-RL | 56.4 | 50.0 | 16.7 | 48.5 | 58.8 | 44.6 | 40.4 | 46.5 |
| | Simple DS | - | - | - | 50.5 | - | - | - | - |
| *ReAct Agentic Frameworks* | | | | | | | | | |
| Qwen-2.5-7B | Vanilla ReAct | 28.2 | 15.3 | 0.0 | 18.4 | 28.1 | 31.2 | 16.0 | 24.2 |
| | **WebDancer** | 41.0 | 30.7 | 0.0 | 31.0 | 40.6 | 44.1 | 28.2 | 36.0 |
| Qwen-2.5-32B | Vanilla ReAct | 46.1 | 26.9 | 0.0 | 31.0 | 35.6 | 38.7 | 22.5 | 31.9 |
| | **WebDancer** | 46.1 | 44.2 | 8.3 | 40.7 | 44.3 | 46.7 | 29.2 | 38.4 |
| QwQ-32B | Vanilla ReAct | 48.7 | 34.6 | 16.6 | 37.8 | 35.6 | 29.1 | 13.2 | 24.1 |
| | **WebDancer** | **61.5** | **50.0** | **25.0** | **51.5** | **52.5** | **59.6** | **35.4** | **47.9** |
| GPT-4o | Vanilla ReAct | 51.2 | 34.6 | 8.3 | 34.6 | 34.6 | 42.0 | 23.9 | 33.8 |

information-seeking and agentic decision-making for these tasks. The closed-source agentic system, *OpenAI DR*, through end-to-end RL training achieves the highest scores. Among Open-sourced frameworks, agentic approaches built on top of native strong reasoning models like QwQ-32B consistently outperform their non-agentic counterparts, demonstrating the effectiveness of leveraging reasoning-specialized models in agent construction. Importantly, under the highly extensible ReAct framework, our proposed **WebDancer** shows substantial gains over the vanilla ReAct baseline across different model scales. Notably, it even surpasses the performance of GPT-4o in the best-case scenario. This demonstrates that even within a lightweight framework, our method significantly enhances agentic capabilities over the underlying base model, validating the strength and generality of our approach. **WebDancer** also

**Results on More Challenging Benchmarks** We evaluate our approach on two more challenging datasets, BrowseComp (*En.*) [13] and BrowseComp-zh (*Zh.*) [20], which are designed to better reflect complex information-seeking scenarios using PASS@1/PASS@3. As shown in Table 2, **WebDancer** demonstrates consistently strong performance across both datasets, highlighting its robustness and effectiveness in handling difficult reasoning and information-seeking tasks.

Table 2: Results on BrowseComp (*En.*) and BrowseComp-zh (*Zh.*).

| Framework | Browsing | *En.* | *Zh.* |
|---|---|---|---|
| GPT-4o | ✗ | 0.6 | 6.2 |
| | ✔ | 1.9 | - |
| QwQ-32B | ✗ | - | 11.1 |
| **WebDancer** | ✔ | 3.8/**7.9** | 18.0/**31.5** |

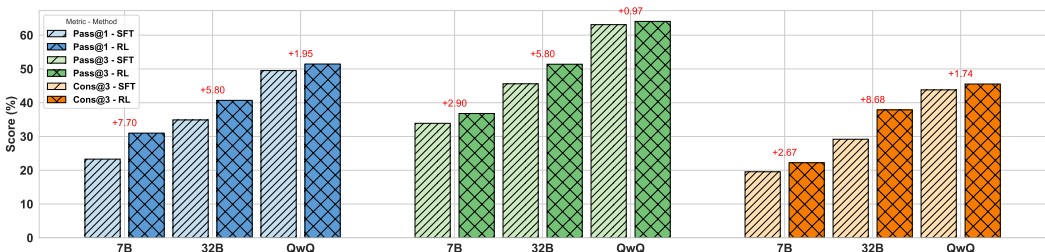

Figure 4: Detailed evaluation results using `Pass@1`, `Pass@3` and `Cons@3` metric on GAIA benchmark.

## 5 Analysis

**Detailed Results** We conduct detailed analyses on the GAIA datasets. Given the dynamic and complex nature of agent environments, as well as the relatively small and variable test set, we further conduct a fine-grained analysis of `Pass@3` and `Cons@3` in Figure 4. The `Cons@3` metric is computed by evaluating the number of correct responses out of three independent attempts: achieving one correct answer yields a score of $1/3$, two correct answers yield $2/3$, and three correct answers result in a full score of $1$. For non-reasoning models, RL leads to substantial improvements in both `Pass@3` and `Cons@3`. Notably, the `Pass@1` performance after RL is comparable to the `Pass@3` of the SFT baseline, consistent with previous findings [37, 38] suggesting that RL can sample correct responses more efficiently. For LRMs, while the improvements in `Pass@1`, `Pass@3`, and `Cons@3` after RL are marginal, a noticeable gain in consistency is observed; this may be due to sparse reward signals caused by excessively long trajectories [39, 40]. This suggests that continued on-policy optimization may yield limited benefits for LRMs in agentic tasks. **Our best-performing model achieves a** `Pass@3` **score of 64.1% on GAIA and 62.0% on WebWalkerQA**.

**High-quality trajectory data is crucial for effective SFT of agents.** We propose two data construction strategies, resulting in the creation of datasets **CRAWLQA** and **E2HQA**. After applying trajectory rejection sampling to the QA data, we further perform filtering to enhance data quality. In Figure 3, we conduct ablation studies on the QwQ and evaluate the effectiveness of the constructed datasets. In long-CoT, hallucinations often arise when the model attempts to answer by simulating observations, primarily due to its exclusive reliance on internal reasoning mechanisms. [5] Final performs better than all under low-data regimes, emphasizing the value of robust filtering.

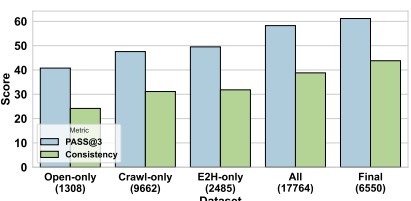

Figure 3: Results on data efficiency using GAIA benchmark. Open-only refers to using only challenging QA datasets from open-source sources.

**SFT for cold start is essential, as the agent tasks demand strong multi-step multi-tool instruction-following capabilities.** We empirically investigate this by comparing performance under a single reinforcement learning setting using QwQ. The results show that the `Pass@3` performance is significantly limited, achieving **only 5%** on the GAIA. For the RL phase, both `Pass@3` and `Cons@3` show consistent improvements as the number of training steps increases, as illustrated in Figure 5a.

Table 3: Results on CoT knowledge transfer. *Inv.* denotes invalid rate. **R.** refers to whether the model is a reasoning model.

| Model | R. | Short-Cot | | | Long-Cot | | |
|---|---|---|---|---|---|---|---|
| | | Pass@3 | Cons@3 | *Inv.* | Pass@3 | Cons@3 | *Inv.* |
| Qwen2.5-7B | ✘ | 33.98 | 22.33 | 0.65% | 35.92 | 21.00 | 21.36% |
| Qwen2.5-32B | ✘ | 42.72 | 24.33 | 4.20% | 45.63 | 30.00 | 13.59% |
| QwQ-32B | ✔ | 44.66 | 28.33 | 0.97% | 58.25 | 39.66 | 13.27% |

**The thinking pattern knowledge used by strong reasoner models is struggle transferable to those of small instruction models.** As shown in Table 3, reasoning models trained on trajectories synthesized by reasoning models significantly enhance their reasoning performance [41]. For non-reasoning models, Long-CoT also demonstrates good performance, but it introduces additional issues, such as a higher invalid rate, often manifested as `repetition`, leading to exceeding the model's context length, particularly in smaller-scale models. These reasoning patterns do not easily transfer to instruction-tuned models, which are generally optimized for task-following behavior rather than deep reasoning. This observation aligns with the findings in [42, 43],

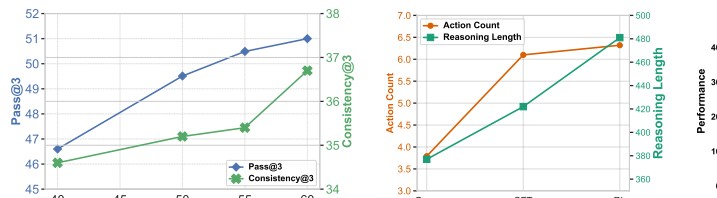
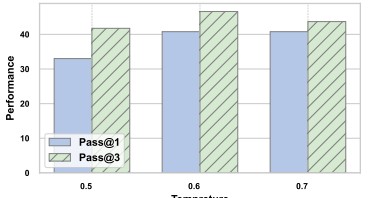

(a) Performance across training steps using the DAPO algorithm.

(b) Evolution of thought length and number of actions.

(c) `Pass@1` and `Pass@3` results on different temperatures.

Figure 5: Analysis on RL algorithm, emergent agency, and agent environments using GAIA benchmark.

which highlight the brittleness of cross-model reasoning knowledge transfer. [4] As such, direct transfer of reasoning capabilities from reasoner models to instruction models remains a non-trivial challenge.

**RL enables longer reasoning processes and supports more complex agentic action.** As demonstrated by the results on Qwen-32B in Figure 5b, we observe that SFT leads to more frequent action generation and extended reasoning sequences, largely due to the nature of our training data (App. E.1). RL frameworks facilitate the emergence of more sophisticated reasoning strategies by allowing models to optimize over sequences of decisions, rather than single-step outputs. This enables models to learn from delayed rewards and engage in deeper exploration of action spaces, leading to more coherent and longer reasoning trajectories. Moreover, RL encourages agentic behaviors where models autonomously decide intermediate steps, subgoals, or tools to achieve final objectives, as shown in App. F. Such capabilities are particularly useful in complex environments where straightforward task-following fails to generalize.

**Web agent executes in a dynamic, evolving environment that inherently resists stabilization.** As shown in Figure 5c, adjusting the decoding temperature had minimal impact on final performance, indicating that decoding variability alone does not account for agent instability. Instead, we attribute much of the performance fluctuation to changes in the web environment itself, highlighting the non-stationary and open-ended nature of real-world agent deployment. Unlike static datasets with fixed distributional properties, real-world environments evolve over time, requiring agents to remain robust under changing contexts and partial observability. Additionally, to further investigate potential overfitting, we conduct a memorization stress test: we fine-tuned a Qwen-7B model on 69 correctly sampled trajectories from the GAIA development set for 10 epochs, and subsequently evaluate its performance on the same set. Despite this, greedy decoding **only achieved 37.4%**, suggesting the difficulty of stabilization on the open-domained agentic tasks.

# 6 Related Works

**Information Seeking Agents and Benchmarks.** Recent advances in information-seeking agents aim to integrate web interaction into LLMs' reasoning. [44] WebThinker [4] and Search-o1 [5] use tool-augmented LLMs that actively retrieve evidence mid-inference. Some works like R1-Searcher [7], ReSearch [6] and Search-R1 [8] focus on reinforcement learning to teach search behavior from outcome-based rewards. DeepResearcher [10] extends this by operating in real web environments with online RL, while SimpleDeepSearcher [9] shows that a small number of distilled demonstrations can train effective agents without full RL. These works demonstrate promising capabilities but often rely on limited or simplistic data. In parallel, benchmarks like GAIA [12] and WebWalkerQA [3] test reasoning and browsing, but many are single-turn or domain-limited. BrowseComp [13] and BrowseComp-zh [20] increase task complexity, requiring multi-hop search and multilingual reasoning, yet still lack diversity and scalability. Our work addresses these gaps by proposing automatic synthesis QA datasets designed to challenge agents across domains and task types in more realistic web environments.

---

[4]We also experiment with mixing short-CoTs and long-CoTs, but observe no significant performance improvements.

**Agents Learning.** Agent learning has evolved from in-context learning towards training-based methods [45–47]. Recent studies [48–50] have primarily focused on leveraging SFT with curated task-solving trajectories following the ReAct paradigm. However, empirical evidence suggests that pure SFT-based agents often exhibit limited generalization performance when confronted with adaptive operational contexts [10, 51–53]. Building upon these limitations, RL-based methods [7, 10, 10, 51, 54] have demonstrated remarkable potential in developing sophisticated search strategies through learned exploration policies. Despite their theoretical advantages, practical implementations face persistent challenges in training stability and sample efficiency. **WebDancer** implements a two-stage framework: an initial cold-start phase employing trajectory-based SFT to establish fundamental agency patterns, followed by targeted RL to cultivate adaptive long-term agency capabilities.

# 7    Conclusion

In this work, we propose a systematic framework for building end-to-end multi-step information-seeking web agents from scratch. By introducing scalable QA data synthesis methods and a two-stage training pipeline combining SFT and on-policy RL, our WebDancer agent achieves strong performance on GAIA and WebWalkerQA. These findings underscore the significance of our proposed training strategy and provide valuable insights into the critical aspects of agent training. Moving forward, this research offers actionable and systematic pathways for the community to advance the development of increasingly sophisticated agentic models capable of tackling complex real-world information-seeking tasks.

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

# A    Limitations

Although our proposed framework has demonstrated promising results, several limitations remain, which point to ongoing efforts and potential directions for future work.

**Tool Number and Type**    Currently, we integrate only two basic information-seeking tools. To enable more advanced and fine-grained retrieval capabilities, we plan to incorporate more sophisticated tools, such as *browser modeling* by abstracting browser functionalities into modular tools, and a *Python* sandbox environment for interacting with external APIs [55–57]. This allows the agent to perform more human-like and efficient interactions, paving the way not only for tackling more challenging benchmarks but also for progressing toward more general and autonomous agency.

**Task Generalization and Benchmarks**    Our current experiments focus on two short-answer information-seeking tasks. However, a comprehensive web agent should also be capable of document-level research and generation [58]. Extending to such open-domain, long-form writing poses significant challenges in *reward modeling* in agentic tasks, which we are actively investigating, particularly how to design more reliable and informative reward signals for long-form generation in open-ended settings [36].

**Data Utilization**    While we have accumulated a large corpus of QA pairs and corresponding trajectories, effectively scaling learning remains a challenge, particularly in the RL stage, where only a small subset (*e.g.*, 5,000 pairs) can be utilized due to computational and stability constraints of RL in agentic tasks. This underscores the need for more efficient data utilization strategies to fully exploit the richness of the collected dataset.

**High Rollout Cost**    The RL phase incurs substantial computational and time overhead, as each rollout involves multiple rounds of tool invocations and LLM completions. This high cost not only limits scalability but also slows down iterative development and experimentation. A promising direction is to develop more efficient mechanisms for integrating tool calls with model completions, which can reduce rollout time and cost without sacrificing learning policy.

**Hybrid Thinking**    We consider two types of datasets characterized by short and long CoTs. Currently, our models are trained on a single dataset type. In future work, we plan to develop a hybrid reasoning agent model capable of dynamically controlling the reasoning length of the agent. [59]

**Thinking Pattern**    In tool invocation, hallucinations may occur. For example, when dealing with mathematical problems, one might erroneously invoke a "*calculate*" tool that does not actually exist. Additionally, over-action may arise during the reasoning process, where redundant actions are performed even after the answer has been confirmed.

# B    Broader Impacts

Building open-source, autonomous web agents capable of long-term information seeking has the potential to greatly benefit scientific research, education, and productivity by democratizing access to complex web-based reasoning tools. However, such systems also raise concerns, including the risk of misinformation propagation if agents rely on unreliable sources, and the possibility of misuse in automated content extraction or surveillance. We emphasize the importance of transparency, source attribution, and responsible deployment practices to mitigate potential harms.

# C    Discussions

## C.1    Concurrent Work

**Comparison with the Training-based Methods**    We primarily compare our approach with two training-based methods: WebThinker and SimpleDeepSearcher, highlighting the key differences. WebThinker also adopts an SFT followed by RL setup, but employs an off-policy RL algorithm [60]. Furthermore, WebThinker triggers actions and observations within the *<thinking_content>*, whereas

our approach adopts a native `ReAct` style architecture, executing each action after completing its corresponding reasoning step. In contrast, Simple DeepSearcher relies solely on supervised fine-tuning over a carefully curated dataset. Our approach similarly follows an SFT-then-RL paradigm, but crucially leverages on-policy RL via DAPO. Our core contribution lies in **building a scalable end-to-end pipeline, from data construction to algorithmic design**, that supports native `ReAct` reasoning. This framework is compatible with both instruction LLMs and LRMs, enabling seamless integration and improved generalization.

**Comparison with the Prompting-based Methods**   Recent efforts in the community have explored building more autonomous and general-purpose agent systems, such as OWL [61, 62], and OpenManus [63], by leveraging foundation models with strong native agentic capabilities, such as Claude [64]. These systems typically rely on carefully engineered agent frameworks and prompting workflows, often involving multi-step tool usage and human-curated task structures. In contrast, we advocate for open-source models with emergent agency, crucial for democratizing agentic AI and advancing fundamental understanding of how agency can arise and scale in open systems. Our native `RAct` framework embraces simplicity, embodying the principle that less is more. **Training native agentic models is fundamentally valuable.**

## C.2   Post-train Agentic Models

Agentic models refer to foundation models that natively support reasoning, decision-making, and multi-step tool use in interactive environments. They exhibit emergent capabilities such as planning, self-reflection, and action execution through structured prompting alone. Recent systems like *DeepSearch* and *Deep Research* illustrate how powerful foundation models can serve as agentic cores, enabling autonomous web interaction through native support for tool invocation and iterative reasoning. However, since web environments are inherently dynamic and partially observable, **reinforcement learning plays a crucial role in improving the agent's adaptability and robustness**. In this work, we aim to elicit autonomous agency in open-source models through targeted post-training.

## C.3   Action Space

In our current design, the agent operates with "search" and "visit" actions, which are considered **fundamental primitives in the information-seeking process** [4, 65]. In principle, these two actions are sufficient to access and retrieve any information available on the web. Moreover, our framework is designed to be modular and extensible. It supports seamless integration with both browser modeling (*e.g.*, scrolling, form filling) and Python sandbox environments, enabling more complex interactions when needed. Given the challenges of sample efficiency in RL settings, we chose to focus on "search" and "visit" as a strong starting point. These tools already demonstrate substantial capabilities across our benchmark tasks.

## C.4   Reward Reliability

In our early-stage experiments, we extensively compared several commonly used reward designs in the search-agent setting, including `recall`, `F1`, and model-based rewards. Our findings motivated the use of LLM-as-Judge. Specifically, both `recall`- and `F1`-based rewards suffered from reward hacking:

- For `recall`, the model often learned to include large numbers of candidate answers in order to boost `recall`, leading to verbose and unreliable outputs.
- For `F1`, the model tended to output only partial answers that scored well but lacked completeness and readability.

As an alternative, we adopt a model-based reward using *LLM-as-Judge*. Our prompts are adapted from established benchmarks (*e.g.*, HLE [28], BrowseComp [13]). To evaluate the robustness of this judge, we tested two strong LLMs: Qwen2.5-72B (open-source) and GPT-4o (closed-source). The results demonstrated **high consistency** between the two. Furthermore, we manually audited 100 samples judged by Qwen2.5-72B and found only one judge error, suggesting strong reliability in practice. While a full quantitative evaluation of judge accuracy remains an interesting direction for

future work, our empirical evidence indicates that the ***LLM-as-Judge*** **is significantly more aligned with the QA task and more robust than standard metric-based rewards in this setting**.

## C.5   Task Generalization

Our current work focuses on short-answer QA tasks, primarily because they offer well-defined evaluation protocols and allow reward computation to be more reliable. In contrast, long-form QA tasks pose significant challenges in both RFT data construction and RL training due to difficulties in evaluation. Interestingly, we observe that our model, trained on short QA tasks, demonstrates strong generalization to long-form settings in terms of information-seeking behavior. This suggests that **the model has learned effective strategies for decomposing complex queries and locating relevant evidence, which naturally transfers to long-form tasks**. We will include a case study to illustrate this generalization behavior in the revised version. However, long-form QA involves not only information retrieval but also generation quality, which remains an open challenge. We plan to explore this direction in future work.

# D   Training Dataset

We collect 40K samples of **E2HQA** and 60K samples of CRAWLQA. These data samples are used to generate trajectories via either QwQ or GPT-4o, followed by a multi-stage filtering process to ensure quality, as described in Sec. 2.2. Table 4 separately reports the statistics for SFT data generated using Long-CoT and Short-CoT reasoning. We plan to scale this high-quality dataset further to investigate whether increasing the data volume leads to significant performance gains in future work.

Table 4: Statistics of training datasets. The thinking length is the average of the tokenized length of the thoughts.

| CoT Type | Num. | Action Count | Thinking Length |
|---|---|---|---|
| **Short** | 7,678 | 4.56 | 510.03 |
| **Long** | 6,550 | 2.31 | 1599.39 |

**Filtering Criterion:** Regarding the trajectory filter employed in Sec. 2.2, it is important to note that, during the quality assessment phase, we mitigate the presence of repetitive patterns by identifying and constraining the maximum occurrence of $n$-grams ($n$=10) within each trajectory to a threshold of 4. The purpose of this is to prevent the model from internalizing detrimental patterns, thereby safeguarding the integrity of the inference process.

**Open-only Datasets:** We select a set of widely-used QA datasets, including MuSiQue [66], Bamboogle [67], PopQA [68], 2Wiki [69], and HotpotQA [70]. To ensure question difficulty, we apply a simple RAG-based filtering process to remove easy questions.

# E   Experimental Details

## E.1   Benchmarks

GAIA is designed to evaluate general AI assistants on complex information retrieval tasks, while WebWalkerQA focuses specifically on deep web information retrieval. Our experiments use 103 questions from GAIA's text-only validation split and 680 questions from the WebWalkerQA test set.

## E.2   Baselines

We compare WebDancer against the following frameworks:

- *No Agency*: which denotes direct use base ability of models and simply uses retrieval-augmented generation (RAG). Includes Qwen2.5-7/32/72B-Instruct [71], QwQ-32B [25], DeepSeek-R1-671B [26], GPT-4o [72].

- *Close-Sourced Agentic Frameworks*: *OpenAI Deep Research (**DR**)* use end-to-end reinforcement learning to complete multitask research tasks.

- *Open-Sourced Agentic Frameworks*: **WebThinker** equips an LRM with a Deep Web Explorer to autonomously search and browse web pages mid-reasoning, interleaving tool use

with chain-of-thought. For a fair comparison, we reproduced the results using Google Search and further replicated both the Base and RL versions of the method. **Search-o1** [5] performs information-seeking by first generating search queries, retrieving web documents, and then using an LLM to answer based on the retrieved content, without optimizing the search process itself. **R1-Searcher** [7] trains an LLM to learn when and how to search using outcome-based reinforcement learning, without any supervised demonstrations.

## E.3 Implements Details

We train using the multi-turn *chatml* format, structuring each dialogue such that tool responses are represented as user messages, and both thoughts and actions generated by the model are represented as assistant messages.

- **Dataset Construction:** The number of reject samplling $N = 5$. The summarizer model $M_s$ is Qwen-2.5-72B. We build our system using the widely adopted `ReAct` framework, implemented on top of the Qwen-Agents [5].
- **Training and Inference:** We construct the judge model $M_j$ based on Qwen-72B-Instruct, and design the reward prompt following [28]. For RL, we implement verl [73, 74] to support the RL algorithm and rollouts. The rollout number in RL is 16. We set the inference parameters as follows: $temperature = 0.6$, $top_p = 0.95$. For the LRM, we use a repetition penalty of 1.1, while for the LLM, the repetition penalty is set to 1.0. In the RL, the temperature of rollout is 1.0 and $top_p = 1.0$.

We conduct all experiments using 32 nodes with 8 NVIDIA H20 (96GB).

## E.4 Prompts for Agent Trajectories Sampling

**Traditional `ReAct` for LLMs**

---

Prompts for `ReAct`

Answer the following questions as best you can.

Use the following format:

Question: the input question you must answer
Thought: you should always think about what to do
Action: the action to take, should be one of [{tool_names}]
Action Input: the input to the action, use JSON Schema with explicit parameters
Observation: the result of the action
... (this Thought/Action/Action Input/Observation can be repeated **many** times)
Thought: you should always think about what to do
Action: Final Answer: the final answer to the original input question

## Execution Framework
1. Thinking phase
- **Mandatory components**:
(a). Evidence chain completeness assessment
(b). Tool selection rationale

2. Action Phase
- **Allowed tools**: Only use tools listed in '{tool_descs}' or can be `Final Answer`, which returns the answer and finishes the task.
You may only provide the 'Final Answer' when you can confidently confirm the answer.
You must also ensure that the 'Final Answer' is accurate and reliable.
To output the Final Answer, use the following template: Final Answer: [YOUR Final Answer]

---

[5] https://github.com/QwenLM/Qwen-Agent/

> 3. Observation phase
> - **Return information from the tool**: The result of the action, you can use the result to think about the next step.
> You have access to the following tools:
>
> {tool_descs}
>
> Begin!
>
> You are likely to use the given tools to gather information and then make the final answer.
> Solve the following question using interleaving thought, action, and observation steps. You may take as many steps as necessary.
> Question: {query}

Figure 6: Prompts for `ReAct` using LLMs.

**Modified `ReAct` for LRMs**

> **Case Trajectory in GAIA**
>
> Answer the following questions as best you can.
> **Allowed tools**: Only use tools listed in {tool_descs}' or can be Final Answer: . You must also ensure that the Final Answer is accurate and reliable.
> You have access to the following tools:
> {tool_descs}
>
> Begin!
> Output Format:
> Action: the action to take, should be one of [{tool_names}] Action Input: the input to the action, use JSON Schema with explicit parameters, when the action is 'Final Answer', do not have Action Input, directly return the answer
> You may take as many steps as necessary. Always use the tools to gather information before making a final answer.
> When you want to make a tool call, please output complete "Action: " and "Action Input: " to make the tool call successful and then output "Observation: " to make the tool call successful.
> Question: {query}

Figure 7: Prompts for `ReAct` using LRMs.

# F  Case Study

As shown in Figure 8, several sophisticated thinking patterns are demonstrated as below:

- **Step-by-step Decomposition** WebDancer breaks down a complex problem into smaller, manageable steps with "First ... Then ... Finally".

- **Hypothesis Testing** WebDancer proposes hypotheses and verifies their validity. It assumes that "Nemo" from Finding Nemo is the orange clownfish (Amphiprion ocellaris). It validates this assumption by consulting the USGS database for non-native occurrences.

- **Handling Information Gaps** When certain information is missing, WebDancer takes action to fill the gap actively.

- **Iterative Reflection** After failing to find the ZIP code in the USGS database, WebDancer realizes it needs to search for the specific address separately. Through a second search, it

successfully retrieves the ZIP code. This demonstrates the Agent's flexibility and adaptability when facing uncertainty.

---

**Case Trajectory in GAIA**

**Question:** I'm researching species that became invasive after people who kept them as pets released them. There's a certain species of fish that was popularized as a pet by being the main character of the movie Finding Nemo. According to the USGS, where was this fish found as a nonnative species, before the year 2020? I need the answer formatted as the five-digit zip codes of the places the species was found, separated by commas if there is more than one place.

**Step 1**

**\<think\>**
Okay, let's tackle this question step by step. The user wants to know the zip codes where the fish from Finding Nemo, which became invasive after being released by pet owners, was found as a non-native species according to USGS before 2020. First, I remember that the main fish in Finding Nemo is Nemo, a clownfish. So the species in question is likely the orange clownfish, Amphiprion ocellaris. But I should confirm that. Maybe I should start by searching for c̈lownfish invasive species USGSẗo see if there's any info. Wait, but the user mentioned the USGS specifically. Let me check what USGS says about ...
**\</think\>**
**\<tool_call\>**
{"name": "search", "arguments": {"query": "USGS Amphiprion ocellaris nonnative locations before 2020"}}
**\</tool_call\>**
**\<tool_response\>**
A Google search for 'USGS Amphiprion ocellaris nonnative locations before 2020' found 9 results:
1. [clown anemonefish (Amphiprion ocellaris) - Species Profile](https://nas.er.usgs.gov/queries/FactSheet.aspx?speciesID=3243) Table 1. States with nonindigenous occurrences, the earliest and ...
2. [clown anemonefish (Amphiprion ocellaris)] (https://nas.er.usgs.gov/viewer/omap.aspx) Citation information: U.S. Geological Survey. [2025]. Nonindigenous Aquatic Species Database. Gainesville, Florida. Accessed [5/9/2025]. Contact us if you are...
...
**\</tool_response\>**

**Step 2**

**\<think\>**
Okay, so I need to find the absolute difference in tens of thousands between the population of chinstrap penguins from the Wikipedia page as of the end of 2018 and the population from the 2020 Nature.com article. The user also wants us to assume two penguins per breeding pair. Let me start by going through the information provided. First, looking at the search results for the Wikipedia part. The sixth result mentions the Chinstrap penguin's conservation status, stating that in 2018, the IUCN estimated around 8 million specimens...
**\</think\>**
**\<tool_call\>**
{"name": "visit", "arguments": {"url": "https://nas.er.usgs.gov/queries/FactSheet.aspx?speciesID=3243", "goal": "Identify states with nonindigenous occurrences of Amphiprion ocellaris before 2020"}}
**\</tool_call\>**
**\<tool_response\>**
{"evidence": "The provided webpage content includes a table listing nonindigenous occurrences of Amphiprion ocellaris. The table shows that one individual was collected in Florida in July 2018 at Fred Howard Park, Pinellas Co., which is before 2...
**\</tool_response\>**

**Step 3**

**\<think\>**

Okay, let's tackle this question step by step. The user is asking about the zip codes where the clownfish (Amphiprion ocellaris), made famous by Finding Nemo, was found as a nonnative species in the US before 2020 according to USGS. First, I did a search using the provided tools. The initial search led me to the USGS Nonindigenous Aquatic Species database page for Amphiprion ocellaris. Visiting that page showed a table listing occurrences. The key point here is that the only entry before 2020 is in Florida, specifically Pinellas County, collected in July 2018 at Fred Howard Park.Now, the next step is converting that location into a zip code ...

**</think>**

**<tool_call>**

{"name": "visit", "arguments": {"url": "https://nas.er.usgs.gov/queries/collectioninfo.aspx?SpeciesID=3243", "goal": "Find specific collection locations with zip codes for Amphiprion ocellaris in the US before 2020"}}

**</tool_call>**

**<tool_response>**

"evidence": "1468741 FL Pinellas Gulf of America, Florida, Fred Howard Park 2018 03100207 Crystal-Pithlachascotee eradicated", "rational": "The webpage provides a collection record for Amphiprion ocellaris (clown anemonefish) in the United States, specifically in Pinellas County, Florida. The record indicates that a specimen was collected at Fred Howard Park in 2018. However, the webpage does not provide zip codes for the collection locations." ...

**</tool_response>**

**Step 4**

**<think>**

So now I need to figure out the zip code for Fred Howard Park in Pinellas County, Florida. Since the USGS data doesn't provide zip codes, I might need to look up the location's address. Let me think if there's another way using the available tools. The allowed tools are search and visit. Maybe I can perform another search to find the zip code of Fred Howard Park. Wait, the user's instructions say to use the tools provided. Since I can't just know the zip code offhand, I should use the search tool again to find the zip code of Fred Howard Park, Pinellas County, Florida. Let me try that. So next action: search for "Fred Howard Park Pinellas County Florida zip code". But I need to structure it properly. Let me check the parameters for the search tool. The query can be anything, so I can input "What is the zip code of Fred Howard Park, Pinellas County, Florida?" After getting that info, I can then present the zip code ...

**</think>**

**<tool_call>**

{"name": "search", "arguments": {"query": "Fred Howard Park Pinellas County Florida zip code"}}

**</tool_call>**

**<tool_response>**

A Google search for 'Fred Howard Park Pinellas County Florida zip code' found 9 results:

1. [Fred Howard Park - Pinellas County](https://pinellas.gov/parks/fred-howard-park/) Fred Howard Park is 155 acres of park and beach located off the Gulf of Mexico in Tarpon Springs. There is a 1-mile causeway from the park to the white sandy ...

2. [Fred Howard Park, 1700 Sunset Dr, Tarpon Springs, FL 34689, US](https://www.mapquest.com/us/florida/fred-howard-park-4792017) ...

**</tool_response>**

**Step 5**

**<think>**

The first link is the species profile for Amphiprion ocellaris, which lists states with nonindigenous occurrences. The summary from visiting that page showed that Florida had a recorded occurrence in 2018 at Fred Howard Park in Pinellas County. Next, I needed to confirm the exact location details. The second visit to the collection info page provided a record from 2018 in Pinellas County, Florida, specifically Fred Howard Park. However, the zip code wasn't listed there, ... , and the specific location's zip code is 34689. Since the question asks for zip codes, and this is the only location mentioned in the provided data,

the answer should be 34689. I need to ensure there aren't other locations, but based on the search results, only Florida's Pinellas County was cited. Thus, the final answer is the zip code 34689.
**</think>**

**<answer>**34689 **<answer>** ✓

Figure 8: Case Study.

