# OpenReview forum: "WebDancer: Towards Autonomous Information Seeking Agency"
_NeurIPS.cc/2025/Conference — NeurIPS 2025 poster_

### Official Review · Reviewer_PfE7 · 2025-06-23

**Clarity:** 3
**Significance:** 2
**Originality:** 2
**Rating:** 4
**Confidence:** 4

**Summary:**

This paper introduces WebDancer, an end-to-end framework for training autonomous web agents to perform complex, multi-step information seeking tasks. The proposed system integrates four stages: (1) QA data generation via CRAWLQA and E2HQA, (2) trajectory sampling using ReAct-style reasoning (Short/Long CoT), (3) supervised fine-tuning (SFT) for cold-start behavior, and (4) reinforcement learning (RL) using the DAPO algorithm for robustness and generalization. WebDancer is evaluated on GAIA, WebWalkerQA, and BrowseComp, achieving strong results that rival or surpass some existing open-source baselines.

**Questions:**

- Line 35 says "unlock the autonomous multi-turn information36 seeking agency", but the interactions seem to be single-turn only, with potentially multiple steps.
- The data quality filtering does not present sufficient details. For example, the heuristic of hallucination detection is based on number of actions, could you explain why?

**Ethical Concerns:**

["NO or VERY MINOR ethics concerns only"]

**Final Justification:**

The rebuttal from the authors addressed some of my concerns about the novelty, hence the score update.

**Limitations:**

Yes

**Quality:**

3

**Strengths And Weaknesses:**

Strengths:
- The paper presents a systematic framework that combines high-quality data synthesis and filtering, SFT, and RL, all within a tool-augmented ReAct agent loop.
- The paper provides detailed studies on the impact of SFT, RL, CoT length, data rejection, and training steps, which reveal rich insights about agent behavior under different setting.
- CRAWLQA and E2HQA introduce diversity and complexity, and rejection sampling and multi-stage filtering ensures high data quality.

Weaknesses:
- The proposed method heavily relies on data distillation from advanced LLMs. Both QA pairs and trajectories are distilled from powerful LLMs (GPT-4o, QwQ), which is a standard practice and limits the novelty.
- Tasks are limited to single-turn QA. The agent solves short-answer queries without any multi-turn interaction, clarification, or open-ended task structure.
- Only two tools (search and click) are used, which limits the tool diversity and task complexity.

---

> ### Author Rebuttal · Authors · 2025-07-31
>
> We sincerely thank you for your insightful comments and constructive suggestions, which are highly valuable for improving our work. Below, we provide point-by-point responses to the raised concerns.
>
>
> ## Q1. Novelty
> **Our core contribution lies in proposing a reusable and end-to-end framework for reproducing a Deep Research agent.**
> This framework consists of four stages: (1) browsing-based data construction, (2) trajectory sampling, (3) supervised fine-tuning for effective cold-start capabilities, and (4) reinforcement learning to enhance generalization.
>
> Throughout this pipeline, we investigate several fundamental questions, such as **how to construct high-quality QA data, how to represent intermediate thoughts, how to design data formats for SFT and RL under deep research scenarios, and how to enable multi-step reasoning**.
> To the best of our knowledge, this is **the first work** that attempts to systematically reproduce Deep Research behavior and provide actionable implementation strategies.
>
> While distillation from advanced LLMs has become a common practice, we would like to clarify that **effective distillation is neither trivial nor devoid of research value**. Our work systematically explores several under-explored but important aspects of this process, including which models to distill from, how to construct and filter high-quality distillation data, and how to best utilize this data for different stages (SFT and RL) of training.
>
> Rather than treating LLM distillation as a plug-and-play component, **we view it as a central research challenge and contribute actionable solutions and insights to make the distillation process effective and scalable in the context of deep research agents.**
>
> ## Q2. Task Generalization
> Thank you for your careful reading and thoughtful comments. We would like to clarify that our current work focuses on **multi-step single-turn QA tasks**. As shown in Appendix F (case study), solving each query often requires the agent to perform **multiple steps of information retrieval, verification, and reasoning**. And **the agent interacts with the user in a single-turn setting**, meaning the user provides only one initial query and does not participate in the intermediate reasoning steps.
>
> We will revise and clarify this point in the main text to avoid misunderstanding.
>
> As for **multi-turn interactions and human-in-the-loop dialogue**, we agree that they are important directions for building more interactive research agents, and we plan to explore them in future work.
>
> Regarding open-ended task structures, our current focus on short-answer QA tasks is due to their well-defined evaluation protocols and relatively reliable reward computation. In contrast, open-ended and long-form tasks present challenges in both RFT data construction and RL training, mainly due to the lack of robust evaluation metrics.
>
> Interestingly, we observe that our model, trained on short QA tasks, **demonstrates strong generalization to long-form settings in terms of information-seeking behavior.** This suggests that the model has learned effective strategies for decomposing complex queries and locating relevant evidence, which naturally transfers to long-form tasks. We will include a case study to illustrate this generalization behavior in the revised version.
>
> We appreciate your suggestion and will consider these broader task settings in future work.
>
> ## Q3: Agent Action Space
>
> We agree that the restricted action space deserves a more explicit discussion. In our current design, the agent operates with "search" and "visit" actions, which are considered **fundamental primitives in the information-seeking process**[1,2].
> In principle, these two actions are sufficient to access and retrieve any information available on the web.
>
> Moreover, our framework is designed to be **modular and extensible**.
> It supports seamless integration with both browser modeling (e.g., scrolling, form filling) and Python sandbox environments, enabling more complex interactions when needed.
> Given the challenges of sample efficiency in RL settings, we chose to focus on "search" and "visit" as a strong starting point.
> These tools already demonstrate substantial capabilities across our benchmark tasks.
>
> We will add further discussion of this design choice and its trade-offs in the revised version.
>
> ## Q4.  Data Quality Filtering
> Thank you for pointing this out. We acknowledge that our current phrasing may be ambiguous. Specifically, the sentence “*we first apply rules to filter out trajectories with more than two actions, ensuring that there are no hallucinations and no severe repetitions*” might be misinterpreted as suggesting that the number of actions alone is used to detect hallucinations.
>
> To clarify: the filtering pipeline operates **on top of** trajectories with more than two actions. We then apply additional filtering steps to ensure quality. In particular:
>
> * **Hallucination detection** is based on inspecting the *thought* steps to check whether the agent generates hallucinated tool outputs (i.e., fabricating a tool return and reasoning based on it, without actually invoking the tool).
> * **Severe repetition** is detected using **n-gram overlap heuristics**, ensuring that repetitive patterns across thoughts and actions are filtered out.
>
> We will revise the manuscript to make this process clearer and avoid confusion. Additionally, we commit to **open-sourcing the full data filtering pipeline**, so that the community can inspect, reproduce, and build upon our approach.
>
> If you have any further questions or suggestions, we would be happy to address them. We kindly ask you to consider our responses when reassessing the paper, and we appreciate your constructive feedback.
>
>
> [1]. OAgents: An Empirical Study of Building Effective Agents
>
> [2]. WebThinker: Empowering Large Reasoning Models with Deep Research Capability

---

> > ### Comment · Reviewer_PfE7 · 2025-08-04
> >
> > Thank the authors for the response. I decide to raise the score.

---

> > > ### Author Response · Authors · 2025-08-04
> > >
> > > Thank you for your follow-up and for raising the score. We truly appreciate your thoughtful evaluation and support!

---

### Official Review · Reviewer_SVnS · 2025-07-01

**Clarity:** 4
**Significance:** 3
**Originality:** 3
**Rating:** 5
**Confidence:** 4

**Summary:**

In this paper, the authors propose WebDancer, an end-to-end framework for building autonomous web agents for multi-step information seeking. The proposed approach consists of four main stages: (1) data synthesis to generate a large corpus of challenging question-answer pairs; (2) high-quality trajectory sampling to generate both short and long CoT reasoning steps; (3) a supervised fine-tuning (SFT) stage to provide the agent with a "cold start" for instruction and tool-use following; and (4) an on-policy RL stage to enhance the agent's decision-making and generalization capabilities. In experiments, extensive results demonstrate the effectiveness of WebDancer on the GAIA and WebWalkerQA benchmarks.

**Questions:**

1. Why do not compare other different RL training methods, such as GRPO?
1. Do the Long-CoT trajectories exhibit more complex planning, self-correction, or deeper exploration, beyond just being longer? Could you provide some cases on Short-CoT and Long-CoT for better understanding and comparison?

**Ethical Concerns:**

["NO or VERY MINOR ethics concerns only"]

**Final Justification:**

My main concerns are addressed.

**Limitations:**

Yes

**Quality:**

3

**Strengths And Weaknesses:**

**Strengths:**

1. This paper addresses a challenging problem of constructing a autonomous  information-seeking agents via a end-to-end paradigm.
1. The proposed four-stage pipeline is comprehensive and systematic. The hybrid training method is practical for achieving efficient agent learning.
1. The experiment design is solid and results show that the proposed method is superior.
1. The paper is well-written, clearly structured and easy to follow.

**Weaknesses:**
1. The training of proposed method is resource-extensive.
1. There may be reward hacking issue, since the evaluation relies on an LLM-as-Judge (72B) for both the RL reward signal and the final performance metric.

---

> ### Author Rebuttal · Authors · 2025-07-31
>
> Thank you very much for your recognition of our work and for the constructive suggestions. We truly appreciate your thoughtful feedback and will address each of your comments in detail below.
>
> ## Q1: Resource-extensive
>
> We acknowledge that our method is resource-intensive during both training and inference. However, our goal is to build a **Deep Research agent** capable of solving high-value problems that often approach or exceed human-level complexity. In such settings, the trade-off between computational cost and reasoning capability is justified.
>
> Notably, even for advanced research agents like those developed by OpenAI, solving a single problem may require **30 minutes to an hour of computation**—a timeframe comparable to what human experts would spend on the same task. We believe that investing computational resources to automate such high-effort, high-reward tasks is both meaningful and impactful for augmenting human productivity.
>
> We will clarify this motivation in the revised version.
>
> ## Q2: : Reward Reliability
>
> In our early-stage experiments, we extensively compared several commonly used reward designs in the search-agent setting, including ***recall***, ***F1***, and ***model-based rewards***. Our findings motivated the use of LLM-as-Judge.
>
> Specifically, both recall- and F1-based rewards suffered from reward hacking:
>
> - For recall, the model often learned to include large numbers of candidate answers in order to boost recall, leading to verbose and unreliable outputs.
>
> - For F1, the model tended to output only partial answers that scored well but lacked completeness and readability.
>
> As an alternative, we adopted a model-based reward using LLM-as-Judge. Our prompts are adapted from established benchmarks (e.g., HLE[3], BrowseComp[4]). To evaluate the robustness of this judge, we tested two strong LLMs: Qwen2.5-72B (open-source) and GPT-4o (closed-source). The results demonstrated **high consistency** between the two.
>
> Furthermore, we manually audited 100 samples judged by Qwen2.5-72B and found only one judge error, **suggesting strong reliability in practice**. While a full quantitative evaluation of judge accuracy remains an interesting direction for future work, our empirical evidence indicates that the LLM-as-Judge is significantly more aligned with the QA task and more robust than standard metric-based rewards in this setting.
>
> We will include a detailed discussion of this aspect in the revised version.
>
> ## Q3: GRPO Performance
> In our early experiments, we also evaluated GRPO on the Qwen-32B backbone and observed that it achieved around 35 points on GAIA, which was lower than DAPO under the same conditions. The performance gap may stem from DAPO’s more effective dynamic sampling of agent trajectories, which better aligns with our task setting.
>
> Given the computational cost and efficiency considerations of RL training, we chose to focus on DAPO as the primary RL algorithm in this work.
>
> ## Q4: Short vs. Long CoT
> Thank you for the insightful question. We agree that simply being "longer" does not necessarily imply better reasoning. In our work, the **Long-CoT trajectories** are not just extended versions of the Short-CoTs; rather, they demonstrate **richer reasoning behaviors**, including:
>
> * **Step-by-step decomposition**
> * **Hypothesis testing**
> * **Handling information gaps**
> * **Iterative reflection and self-correction**
>
> As illustrated in **Appendix F**, Long-CoTs often break down complex queries into subgoals, assess intermediate evidence, and revise earlier assumptions based on new findings. In contrast, **Short-CoTs**, while still logical, tend to follow a more direct and surface-level path with limited internal deliberation.
>
> For example, in response to a question about the presence of a nonnative species (clownfish) in the US before 2020, a **Short-CoT** may look like this (with tool outputs omitted for brevity):
>
> > *To find out where this species was found as a nonnative species in the United States before 2020, I will need to search for information from the USGS or related sources that track invasive species. I will start by searching for the clownfish and its status as an invasive species in the US.*
>
> > *The search results provide several resources related to the clownfish and its status as an invasive species. The most promising sources seem to be from the USGS database, which specifically tracks nonindigenous aquatic species. I should visit the relevant USGS webpage to gather detailed information about clownfish occurrences in the United States as a nonnative species.*
>
> > *The Wikipedia page did not provide any additional information regarding clownfish being found as nonnative species in the US before 2020.*
>
> While this Short-CoT is coherent and goal-oriented, it **lacks deeper exploration**, such as verifying conflicting sources or revisiting assumptions when evidence is missing.
>
> We will **include a side-by-side comparison between Short-CoT and Long-CoT** examples in the revised version to better illustrate these distinctions.
>
> Once again, we sincerely thank you for your recognition of our work and your thoughtful feedback. If you have any further questions or suggestions, we would be more than happy to address them.

---

> > ### Comment · Reviewer_SVnS · 2025-08-03
> >
> > Thank you for your rebuttal. My main concerns are addressed.

---

> ### Author Response · Authors · 2025-08-04
>
> Thank you for your follow-up. We’re glad our rebuttal addressed your main concerns. We appreciate your thoughtful review and recognition.

---

### Official Review · Reviewer_Q7WQ · 2025-07-03

**Clarity:** 2
**Significance:** 3
**Originality:** 2
**Rating:** 4
**Confidence:** 3

**Summary:**

This paper presents a pipeline for training information seeking agents that can answer closed-form answers through multi-step interactions with the web environment.
Specifically, the proposed pipeline involves four steps: 1) collecting high-quality QA pairs for training; 2) sample trajectories using both LLMs and LRMs for QA pairs; 3) performing SFT on the filtered trajectories; and 4) performing RL on the QA pairs for enhanced generalization.
Along with the pipeline, this paper collects two large-scale QA pairs for training, i.e., CrawlQA (60K) and E2HQA (40K).
This paper applies the training pipeline to fine-tune Qwen-2.5-7B, Qwen-2.5-32B, and QwQ-32B. Experimental results demonstrate the effectiveness of the training pipeline on both GAIA and WebWalkerQA.

**Questions:**

1. Do you plan to release CrawlQA and E2HQA to the public, including open-source the code for data curation?
2. In CrawlQA, how to make sure the navigation is meaningful and does not have a corresponding shortcut? For example, you may get the answer page after 10 clicks, but there might be a shortcut to directly go there or that page might be indexed by google search already.
3. In addition to the dataset and trained models, what major insights do you think are important to convey to readers?

Clarification:
a. Is it possible that for the same question, there's both a short CoT version trajectory and a long CoT version? If so, would combining them for training result in some conflicts?
b. Based on appendix D, only less than 10% questions were retained during RFT?

**Ethical Concerns:**

["NO or VERY MINOR ethics concerns only"]

**Final Justification:**

I decide to raise my overall evaluation after reading the authors' response.

**Limitations:**

1. The most important use case of information seeking agents typically extend beyond closed-form QA. For example, there can be more open-ended needs such as finding a products with certain criteria or even generating a report for a science question. It might be an interesting future direction. That said, I agree this is not within the scope of this paper. So this is just a general suggestion.
2. This paper may reveal more details regarding the constructed dataset for people to better assess it.

**Quality:**

2

**Strengths And Weaknesses:**

### Strengths:
1. The curated QA pairs, in particular CrawlQA, can be a valuable source of verifiable rewards for training information seeking agents.
2. The experiments are comprehensive. Necessary ablations are conducted to justify the designs.

### Weaknesses:
1. Important details regarding the dataset construction are not revealed in this paper, which might make it difficult to assess the quality of the work. This is important because the techniques used for training in this paper are mostly existing ones, while the dataset is the crucial differentiating point.
2. Related to the first point, the space arrangement of this paper could be improved. Currently, it spend majority of the space on the training techniques, while things like ReAct, GRPO, or the reward design with both format and correctness are something people already familiar with.
3. (Minor) This paper may benefit from polished writing. Some sections appear to be verbose (e.g., Sec 2). In addition, there are several minor issues: a) Figure 1 seems not referred in the main text? b) line 274 is struggle transferable, grammar error

---

> ### Author Rebuttal · Authors · 2025-07-31
>
> We sincerely thank you for your insightful comments and constructive suggestions, which are highly valuable for improving our work. Below, we provide point-by-point responses to the raised concerns.
>
> ## Q1. Dataset Open-Sourcing
> **We are committed to open-sourcing all code used for data generation and curation.** Furthermore, we will release the full CrawlQA and E2HQA datasets on open-access platforms such as HuggingFace and ModelScope. By doing so, we aim to contribute to and support the broader open-source research community.
>
> ## Q2. Quality of CrawlQA
> In CrawlQA, our construction strategy follows the construction methodology introduced in WebWalkerQA[1], which demonstrates **that information buried deep within websites is generally not easily accessible via search engines**.
> However, we acknowledge that some queries may still have shortcuts due to search engine indexing.
> To address this, we adopt a filtering step inspired by BrowseComp[2]: we employ a basic agent built on Qwen-72B with search capabilities to execute. **Queries that can be resolved in a single-step search are removed.**
> This ensures that the retained QA pairs require meaningful multi-step navigation and better reflect realistic web exploration challenges.
> We will include further details about this filtering process in the revised version.
>
> ## Q3. Space Arrangement and Writing
> Our core contribution lies in presenting a systematic, end-to-end, and actionable reproduction of a Deep Research agent.
> Thanks for your advice.
> Given the complexity of the full pipeline, we will allocate substantial space to ensure clarity and completeness.
> That said, we will revise the space arrangement to better balance novelty and background content, and refine the writing to be more concise and focused.
>
> ## Q4. Insights and Novelty
> **Our core contribution lies in proposing a reusable and end-to-end framework for reproducing a Deep Research agent.**
> This framework consists of four stages: (1) browsing-based data construction, (2) trajectory sampling, (3) supervised fine-tuning for effective cold-start capabilities, and (4) reinforcement learning to enhance generalization.
>
> Throughout this pipeline, we investigate several fundamental questions, such as **how to construct high-quality QA data, how to represent intermediate thoughts, how to design data formats for SFT and RL under deep research scenarios, and how to enable multi-step reasoning**.
> To the best of our knowledge, this is **the first work** that attempts to systematically reproduce Deep Research behavior and provide actionable implementation strategies.
>
> ## Q5. CoT Styles and Filtering Efficiency
> For each query, we can collect both a short CoT and a long CoT version of the reasoning trajectory.
> Importantly, we train the two styles separately to avoid interference: as shown in Table 3, the instruct model is trained with short CoTs, while the reasoning model, like QwQ, uses long CoTs.
>
> We also conducted transfer experiments across styles, and the results support the effectiveness of our current setup.
> Additionally, we experimented with combining both CoT styles during training, but as you anticipated, this led to performance degradation due to conflicting reasoning patterns between short and long CoTs.
> This confirms that treating them separately is a more effective strategy.
>
> Regarding the filtering efficiency, for queries used in RFT training, we apply multiple layers of filtering, including *format validity* control during rejection sampling, *correctness verification* of final answers, *quality assessment* of thought steps, and constraints on *the number of actions taken*.
>
> **Our QA construction pipeline is designed to be scalable**: as long as high-quality RFT-style trajectories can be generated, the corresponding QA examples are meaningful and useful. While high-quality QA data with detailed thoughts is crucial for RFT, other QA instances without such strict requirements can still be effectively used in the RL stage.
>
> We will supplement and clarify this part in the revised version.
>
> ## Q6. Task Generalization
> Our current work focuses on short-answer QA tasks, primarily because they offer well-defined evaluation protocols and allow reward computation to be more reliable. In contrast, long-form QA tasks pose significant challenges in both RFT data construction and RL training due to difficulties in evaluation.
>
> Interestingly, we observe that our model, trained on short QA tasks, **demonstrates strong generalization to long-form settings in terms of information-seeking behavior**. This suggests that the model has learned effective strategies for decomposing complex queries and locating relevant evidence, which naturally transfers to long-form tasks. We will include a case study to illustrate this generalization behavior in the revised version.
>
> However, long-form QA involves not only information retrieval but also generation quality, which remains an open challenge. We plan to explore this direction in future work.
>
> If you have any further questions or suggestions, we would be happy to address them. We kindly ask you to consider our responses when reassessing the paper, and we appreciate your constructive feedback.
>
> [1]. WebWalker: Benchmarking LLMs in Web Traversal
>
> [2]. BrowseComp: A Simple Yet Challenging Benchmark for Browsing Agents

---

> ### Author Response · Authors · 2025-08-08
>
> Dear Reviewer Q7WQ,
>
> We would like to kindly remind you that the discussion period is **approaching its end, with approximately 24 hours remaining**. We truly appreciate the thoughtful feedback you have already provided and wanted to check if you might have any follow-up thoughts or questions regarding our rebuttal.
>
> We are more than happy to **further clarify any remaining concerns or provide additional details to support a productive discussion**. Your continued input would be greatly valued as part of this collaborative review process.
>
> Thank you again for your time!

---

### Official Review · Reviewer_4wHM · 2025-07-06

**Clarity:** 2
**Significance:** 3
**Originality:** 3
**Rating:** 4
**Confidence:** 4

**Summary:**

This paper presents WebDancer, a systematic and scalable framework for training web-based autonomous information-seeking agents. The framework follows a four-stage pipeline: (1) constructing diverse and challenging QA datasets (CRAWLQA and E2HQA), (2) generating agentic trajectories using rejection sampling under both short and long Chain-of-Thought (CoT) prompting strategies, (3) supervised fine-tuning (SFT) to provide a strong initialization, and (4) reinforcement learning (RL) with the DAPO algorithm to improve generalization. WebDancer is instantiated as a ReAct-style web agent, and evaluated on two challenging benchmarks (GAIA and WebWalkerQA), demonstrating strong performance compared to both open-source and closed-source agentic frameworks.

**Questions:**

See Weaknesses part

**Ethical Concerns:**

["NO or VERY MINOR ethics concerns only"]

**Final Justification:**

I thank the authors for the responses. I maintain my initial positive rating of this paper.

**Limitations:**

yes

**Paper Formatting Concerns:**

No Formatting Concerns

**Quality:**

3

**Strengths And Weaknesses:**

**Strengths**

1. The direction of building autonomous information-seeking agents is both timely and important for advancing agentic AI capabilities.

2. The proposed four-stage framework (data construction, trajectory sampling, SFT, RL) is clear and well-structured.

**Weaknesses**

1. The paper would benefit from improved exposition. Some essential context is missing, such as a basic introduction to the unique characteristics and challenges of web agent tasks for readers not already familiar with this area. Adding more illustrative case studies could further enhance readability and practical relevance.
2. The experiments focus primarily on the Qwen series as backbone models. It remains unclear how generalizable the proposed pipeline is to other LLM backbones.
3. The agent’s action space is restricted to “search” and “click/visit,” with no support for more complex interactions such as scrolling or system key events. This restriction should be discussed more explicitly.
4. The RL reward relies on LLM-as-Judge, but the paper does not provide an explicit evaluation of this judge’s accuracy. Since the judge could become a bottleneck, an analysis or discussion of its reliability is essential.
5. Could the authors provide more detailed case studies highlighting which types of web tasks or interaction patterns benefit the most from the SFT or RL stages? Conversely, what are the common failure modes or limitations in the current system, and how might they be addressed?

---

> ### Author Rebuttal · Authors · 2025-07-30
>
> We sincerely thank you for your insightful comments and constructive suggestions, which are highly valuable for improving our work. Below, we provide point-by-point responses to the raised concerns.
> ## Q1: Exposition Improvement
> Search-based web agent tasks present a set of unique challenges that differentiate them from traditional web-based QA or single-turn retrieval tasks. Specifically:
>
> 1. **Lack of High-Quality, High-Difficulty Training Data**:
> Unlike simpler retrieval tasks, search-agent benchmarks such as *Gaia* and *BrowserComp* involve complex multi-hop reasoning and multi-step interaction. However, there is a significant scarcity of training data that reflects this level of difficulty. The few existing training datasets are typically not well-aligned with the test distributions of these benchmarks. These benchmarks feature long-horizon tasks, which require capabilities beyond what standard SFT data can teach.
>
> 2. **Limited Support for Multi-Step Search and Deep Research**:
> Current agent models often struggle with multi-step retrieval and iterative refinement. This limitation hampers their ability to perform *deep research*, i.e., cases where information must be verified, cross-checked, or gathered from multiple heterogeneous sources.
>
> 4. **Lack of Systematic Guidance for Deep Research Behavior**:
> Before our work, there has been no systematic methodology for training or evaluating agents on “deep research” tasks. Most existing approaches focus on shallow interactions or fail to generalize to tasks requiring flexible search strategies, adaptation, and reflection.
>
> Our work aims to address these gaps by providing both a practical framework and empirical insights into training agents capable of sophisticated web-based research behaviors.
>
> We will enhance the Introduction and Related Works sections with a clearer and more comprehensive introduction to the unique characteristics and challenges of web agent tasks, to better support readers who may not be familiar with this area. Additionally, while we have already included a case study in Appendix F, we appreciate the suggestion to make the practical relevance more prominent. We will incorporate a more illustrative and motivating case study directly in the Introduction to help guide the reader and highlight the real-world significance of our setting.
>
> ## Q2: Backbone Generalization
>
> Due to computational constraints, we primarily conducted experiments using the Qwen2.5 series as backbone models.
> We selected Qwen2.5 for its competitive performance and open availability, which allowed for a thorough and reproducible evaluation of our proposed pipeline.
> Importantly, **we have validated our approach across multiple sizes of Qwen2.5 (e.g., 7B, 32B), consistently observing performance improvements**, which supports the generality of our method across model scales.
> In future work, we plan to extend our study to other backbone families, such as LLaMA and Mixtral, to further verify the robustness and generalizability of our approach across diverse architectures.
>
> ## Q3: Agent Action Space
>
> We agree that the restricted action space deserves a more explicit discussion. In our current design, the agent operates with "search" and "visit" actions, which are considered **fundamental primitives in the information-seeking process**[1,2].
> In principle, these two actions are sufficient to access and retrieve any information available on the web.
>
> Moreover, our framework is designed to be **modular and extensible**.
> It supports seamless integration with both browser modeling (e.g., scrolling, form filling) and Python sandbox environments, enabling more complex interactions when needed.
> Given the challenges of sample efficiency in RL settings, we chose to focus on "search" and "visit" as a strong starting point.
> These tools already demonstrate substantial capabilities across our benchmark tasks.
>
> We will add further discussion of this design choice and its trade-offs in the revised version.
>
> ## Q4: Reward Reliability
>
> In our early-stage experiments, we extensively compared several commonly used reward designs in the search-agent setting, including ***recall***, ***F1***, and ***model-based rewards***. Our findings motivated the use of LLM-as-Judge.
>
> Specifically, both recall- and F1-based rewards suffered from reward hacking:
>
> - For recall, the model often learned to include large numbers of candidate answers in order to boost recall, leading to verbose and unreliable outputs.
>
> - For F1, the model tended to output only partial answers that scored well but lacked completeness and readability.
>
> As an alternative, we adopted a model-based reward using LLM-as-Judge. Our prompts are adapted from established benchmarks (e.g., HLE[3], BrowseComp[4]). To evaluate the robustness of this judge, we tested two strong LLMs: Qwen2.5-72B (open-source) and GPT-4o (closed-source). The results demonstrated **high consistency** between the two.
>
> Furthermore, we manually audited 100 samples judged by Qwen2.5-72B and found only one judge error, **suggesting strong reliability in practice**. While a full quantitative evaluation of judge accuracy remains an interesting direction for future work, our empirical evidence indicates that the LLM-as-Judge is significantly more aligned with the QA task and more robust than standard metric-based rewards in this setting.
>
> We will include a detailed discussion of this aspect in the revised version.
> ## Q5: Interaction Patterns and Agent Behavior
> **Current agent models lack robust multi-step reasoning and iterative retrieval, limiting their effectiveness in deep research tasks requiring cross-verification and multi-source synthesis.**
> As shown in Appendix F, we have conducted a case-based analysis of agent behaviors in the GAIS task. We agree that a more detailed analysis of interaction patterns is valuable. Our current findings indicate that:
>
> - In the SFT stage, the model already learns **effective decomposition and high-level planning patterns**, such as structured sequences like *"First ... Then ... Finally ..."*.
>
> - In the RL stage, the agent further acquires more advanced behaviors such as **cross-validation, reflection, and adaptive replanning**. For the case in Appendix F, in a complex task involving ZIP code lookup, the agent first fails to find the result via the USGS database, but then adapts by issuing a second targeted search, ultimately succeeding, demonstrating robust handling of uncertainty.
>
> Regarding failure modes and current limitations, we acknowledge that longer interaction turns introduce long-context challenges in our proposed WebDancer, which can affect coherence and memory. As part of future work, we are exploring Long-CoT to Short-CoT condensation and summarization techniques to mitigate these issues.
>
> We will expand this discussion and include additional case studies in the revised version to further clarify the evolution of interaction strategies and the impact of training on agent sophistication.
>
> We sincerely appreciate your valuable suggestions, which are highly helpful for improving our work. We will incorporate all of them in the revised version. Please feel free to raise any additional concerns.
>
> [1]. OAgents: An Empirical Study of Building Effective Agents
>
> [2]. WebThinker: Empowering Large Reasoning Models with Deep Research Capability
>
> [3]. Humanity's Last Exam
>
> [4]. BrowseComp: A Simple Yet Challenging Benchmark for Browsing Agents

---

> > ### Comment · Reviewer_4wHM · 2025-08-05
> >
> > I thank the authors for the responses. I maintain my initial positive rating of this paper.

---

> > > ### Author Response · Authors · 2025-08-05
> > >
> > > We sincerely thank you for your positive evaluation and continued support of our work. We greatly appreciate your time and effort in reviewing our paper and your encouraging feedback.

---

### Comment · Area_Chair_KqRy · 2025-08-03
**Please discuss the paper asap**

Dear reviewers,

Now the rebuttal is available. Thanks 4Wjo for starting the discussion. Others, please discuss with authors and among reviewers asap.

Please try to come to a consensus on the key issues even though the rating can be different. Please feel free to let me know how I can help.

Best,

Your AC

---

### Author Response · Authors · 2025-08-09
**Summary of Contributions and Discussions**

Dear Reviewers and ACs/SACs,

We sincerely thank the discussion participants and all reviewers for their time and constructive feedback. Our final ratings may have been `5, 4, 4, and 3` (exact numbers hidden after the final rating). For the 3 score, although **there was no interaction throughout the entire discussion process and the confidence was low**, we believe we **have addressed the concerns and received clear recognition from other reviewers**. On the aspects of novelty and task generalization, we were encouraged by the positive assessments from several reviewers. For concerns about writing and open-sourcing, we will make targeted improvements in the revised version.

Below, we summarize our key contributions:

- Reusable End-to-End Framework for Deep Research Agents
We propose a systematic framework to reproduce a Deep Research agent, structured into four stages: (1) Browsing-based data construction, (2) Trajectory sampling, (3) Supervised fine-tuning (SFT) for effective cold-start capabilities, (4) Reinforcement learning (RL) to enhance generalization.

- Throughout this pipeline, we investigate several fundamental questions, such as how to construct high-quality QA data, how to represent intermediate thoughts, how to design data formats for SFT and RL under deep research scenarios, and how to enable multi-step reasoning. To the best of our knowledge, **this is the first work that attempts to systematically reproduce Deep Research behavior and provide actionable implementation strategies.**

Addressing Key Concerns and Planned Revisions:

- **Open Source Commitment**: We will release the data and the data preprocessing pipeline.

- **Case Study**: We will compare the effectiveness of short vs. long chain-of-thought (CoT) reasoning.

- **Additional Discussion**: We will expand the discussion on reward reliability and task generalization.

Given the strong positive feedback from multiple reviewers, we believe our work makes a meaningful contribution to advancing reproducible, multi-stage Deep Research systems, and we remain committed to refining it further in the revised version.

---

### Decision · Program_Chairs · 2025-09-17

**Decision:**

Accept (poster)

**Comment:**

**Recommendation:** Need Discussion

### (a) Summary of the Paper

This paper introduces WebDancer, a comprehensive, four-stage framework for building autonomous information-seeking web agents capable of "deep research." The proposed pipeline consists of: (1) constructing new, challenging QA datasets from web data, (2) sampling high-quality agentic trajectories using LLMs, (3) using these trajectories for supervised fine-tuning (SFT) to establish a strong baseline agent, and (4) further improving the agent's generalization and decision-making capabilities through reinforcement learning (RL). The authors instantiate this framework to train Qwen-based agents and demonstrate strong performance on challenging benchmarks like GAIA and WebWalkerQA.

The paper received a positive reception from all reviewers, with final scores leaning toward acceptance (5, 4, 4, 4). The authors provided a thorough and effective rebuttal, addressing most of the initial concerns and leading multiple reviewers to raise their scores. Despite this positive consensus on the paper's quality and empirical strength, a "Need Discussion" is warranted to deliberate on the nature and significance of the contribution, particularly regarding its novelty.

### (b) The Case for Acceptance

The primary argument for acceptance, shared by all reviewers, is that the paper presents a **valuable, systematic, and reproducible framework** for a highly relevant and challenging problem. Instead of proposing a single algorithmic tweak, the authors provide a complete, end-to-end blueprint for creating a sophisticated web agent. This systems-level contribution is a significant strength. Key points in favor of acceptance include:

*   **A Comprehensive and Actionable Pipeline:** The four-stage process is well-structured, logical, and provides clear, actionable guidance for researchers looking to build similar "deep research" agents. Reviewers praised this as a "comprehensive and systematic" approach (SVnS) and a "clear and well-structured" framework (4wHM).
*   **Strong Empirical Results:** The paper demonstrates impressive performance on difficult, multi-step information-seeking benchmarks, validating the efficacy of the proposed pipeline. The solid experimental design and thorough ablations were also commended.
*   **Significant Data Contribution:** The creation and planned release of two new large-scale QA datasets, CrawlQA and E2HQA, is a valuable contribution to the community in its own right, addressing a known scarcity of high-quality training data for this task (Q7WQ).
*   **Effective Rebuttal:** The authors successfully addressed numerous concerns during the rebuttal, providing clarifications on their data filtering process, validating the reliability of their LLM-as-Judge through a manual audit, and committing to open-sourcing all data and code. This responsiveness strengthened the paper and convinced several reviewers to raise their scores.

### (c) The Case for a More Cautious Stance

While empirically strong, the paper's primary weakness, noted by multiple reviewers (PfE7, Q7WQ), is its **limited conceptual novelty**. The contribution lies in the effective integration and systematization of existing techniques, rather than the invention of new ones. This raises a valid question about the paper's fit for NeurIPS. Key points for a more critical evaluation include:

*   **Reliance on Existing Components:** The core components of the pipeline—data distillation from powerful LLMs, the ReAct agent format, SFT for cold-starting, and on-policy RL algorithms like DAPO, are all established techniques in the agent literature.
*   **Data-Centric Focus:** A significant portion of the paper's contribution is data-centric (the new datasets and the trajectory sampling process). While valuable, this positions the work more as an empirical or resource paper, which can sometimes be seen as less impactful than a work with fundamental methodological advances.
*   **Scope Limitations:** The agent's action space is limited (primarily search and click), and the tasks are constrained to single-turn QA. While the authors provide reasonable justifications, this narrows the scope of the demonstrated capabilities.

### (d) Key Points for AC Discussion

The consensus is that this is a high-quality empirical paper that provides a valuable service to the community. The core point of discussion is whether this type of contribution, a systematic, reproducible, and highly effective integration of existing methods for a complex task—meets the novelty and impact bar for NeurIPS. The discussion points are:

1.  **Nature of Contribution: Systematization vs. Algorithmic Novelty:** How should we weigh a strong, novel *system* against a work with more fundamental algorithmic novelty? This paper provides a clear blueprint for reproducing a complex agentic behavior. Is this "how-to" guide, backed by strong results, a sufficiently impactful contribution for the applications track, even if the individual components are not new?

2.  **Evaluating the Data Contribution:** The paper's new datasets are a significant asset. How much weight should be given to this resource contribution in the final decision? Reviewer Q7WQ initially pointed out a lack of detail on the data construction, which the authors addressed in the rebuttal. Does the promise to release the data and the full pipeline sufficiently validate this aspect of the work?

3.  **Reliability of the Evaluation:** The RL stage relies on an LLM-as-Judge. While the authors provided a manual audit in the rebuttal showing high reliability, this was not part of the original submission. Is this post-hoc validation sufficient to alleviate concerns about potential reward hacking or evaluation bias in a core component of the training pipeline?

Overall, this is a strong paper that the community would likely find useful. The discussion should center on whether its strengths as a practical, reproducible framework and a valuable data resource are sufficient to overcome the concerns about its limited originality in terms of core techniques.